# Dynamics of primary productivity in the northeastern Bay of Bengal over the last 26,000 years

3 Xinquan Zhou<sup>1</sup>, Stéphanie Duchamp-Alphonse<sup>1</sup>, Masa Kageyama<sup>2</sup>, Franck Bassinot<sup>2</sup>, Luc
4 Beaufort<sup>3</sup>, Christophe Colin<sup>1</sup>

<sup>1</sup>Université Paris-Saclay, Géosciences Paris Sud, UMR 8148, CNRS, Rue du Belvédère, 91405 Orsay,
France

<sup>2</sup>Laboratoire des Sciences du Climat et de l'Environnement, UMR 8112, CEA/CNRS/UVSQ, Université

Paris-Saclay, Centre CEA-Saclay, Orme des Merisiers, 91191 Gif-sur-Yvette, France

<sup>3</sup>Centre de Recherche et d'Enseignement de Géosciences de l'Environnement, UMR 7330,

CNRS/IRD/Aix-Marseille Université, Av. Louis Philibert, BP80, 13545, Aix-en-Provence, France

Correspondance to: Xinquan Zhou (xinquan.zhou@universite-paris-saclay.fr)

Abstract. At present, variations of primary productivity (PP) in the Bay of Bengal (BoB) are driven by salinityrelated-stratification, which is controlled by the Indian Summer Monsoon (ISM). The relationships between PP, 13 precipitation, and more generally climate in the past, are not clearly understood. Here, we present a new record of 14 PP based on the examination of coccolithophore assemblages in a 26,000-year sedimentary series, retrieved in the 15 northeastern BoB (core MD77-176). We compare our PP records to published climate and monsoon records, as 16 17 well as outputs from numerical experiments obtained with the Earth System Model IPSL-CM5A-LR, including 18 marine biogeochemical component PISCES, and with the transient climate simulation TraCE-21. Our results show that PP was most probably controlled by nutrient contents and distribution within the upper water column, that 19 were predominantly influenced by (i) regional river systems between 26 and 19 kyr BP, i.e. when sea-level was 20 relatively low, and climate was relatively dry and (ii) salinity-related stratification over the last 19 kyr BP, i.e. when 21 22 sea level rose and more humid conditions prevailed. During that period, salinity and stratification were directly related to monsoon precipitation dynamics, which were chiefly forced by both insolation and AMOC strength. 23 During Heinrich Stadial 1 and Younger Dryas, i.e. when the AMOC collapsed, weaker South Asian precipitation 24 diminished stratification and enhanced PP. During Bølling-Allerød, i.e. when the AMOC recovered, stronger South 25 26 Asian precipitation increased stratification and subdued PP. Similarly, the precipitation peak recorded around the 27 middle-early Holocene is consistent with a stronger stratification that drives PP minima.

# 28 1. Introduction

The climatology and oceanography of South Asia and the North Indian Ocean are dominated by the Indian 30 Monsoon, which is characterized by strong seasonal contrasts in wind and precipitation patterns (Shankar et al., 31 2002; Gadgil et al., 2003). During the northern hemisphere summer season, the North Indian Ocean is strongly 32 influenced by the southwesterly winds blowing from the sea toward the Asian continent, thus carrying large amounts of moisture on land. During the winter season, the winds blow over the continent toward the Indian Ocean 33 from the northeast, thus causing relatively dry conditions on land, with precipitations moved over the ocean. 34 35 Monsoon precipitations are directly associated to the position of the Inter Tropical Convergence Zone (ITCZ; Schneider et al., 2014), whose latitudinal displacement is paced by seasonal changes in insolation due to the 36 obliquity of the Earth's axis and precession, and results from variations in the land-sea thermal contrast caused by 37 differences of heat capacity of the continent and the ocean (Meehl, 1994, 1997; Webster, 1998; Wang et al., 2003). 38 39 It is also influenced by teleconnections with the El Niño-Southern Oscillation and the Indian Ocean Dipole, two 40 climate modes of interannual variability that develop from air-sea interactions in the tropical Pacific, and drive significant changes within the Indian Ocean (Ashok et al., 2004; Wang et al., 2008; Currie et al., 2013; Jourdain et 41 42 al., 2013).

Remarkably, the eastern part of the North Indian Ocean, i.e. the Bay of Bengal (BoB) and the Andaman Sea, receives heavier annual precipitation than its western counterpart, i.e. the Arabian Sea (AS). This pattern, together 44 45 with differences in local evaporation result in hydrological and ecological differences between these two areas (e.g. Prasanna Kumar et al., 2002; Vinayachandran et al., 2002; Shenoi et al., 2002; Shi W et al., 2002; Dey and Singh, 46 2003; Rao and Sivakumar, 2003; Prasad, 2004; Currie et al., 2013). A noteworthy characteristic of modern 47 conditions prevailing in the North Indian Ocean is the low PP in the BoB and the Andaman Sea compared to the 48 AS (Prasanna Kumar et al., 2002). Previous studies revealed that low annual PP in the BoB results from the 49 50 important freshwater input by rivers and direct rainfall on the sea, which cause a strong stratification of the upper seawater column and an impoverishment of nutrients in surface layers (Vinayachandran et al., 2002; Madhupratap 51 et al., 2003; Gauns et al., 2005). In contrast, the AS has a high PP which is mainly associated with high nutrient 52 content in the upper layer, thanks to wind driven mixing during winter and coastal upwelling during summer (Schott, 53 1983; Anderson and Prell, 1992; Madhupratap et al., 1996; Gardner et al., 1999; Prasanna Kumar et al., 2001; 54 55 Wiggert et al., 2005). Both, the BoB and AS are characterized by a relatively small sea surface temperature (SST) seasonal cycle. Thus, seasonal and interannual changes in PP result chiefly from variations in the nutricline depth 56 (i.e. variations in nutrient availability in the photic zone) controlled by salinity-related stratification of the upper 57 58 seawater column in relation to local evaporation-precipitation balance and river runoff, and/or dynamical processes

such as wind-driven mixing/upwelling (e.g. Lévy et al., 2001; Vinayachandran et al., 2002; Chiba et al., 2004; Rao
et al., 2011; van de Poll et al., 2013; Behara and Vinayachandran, 2016; Spiro Haeger and Mahadevan, 2018).

Past changes of PP at both orbital- and millennial-scales in the western and northern AS have been widely 61 62 studied, and authors have interpreted PP variations as chiefly reflecting changes in the intensity of Indian Summer 63 Monsoon (ISM) southwesterly winds (e.g. Schultz et al., 1998; Ivanova et al., 2003; Ivanochko et al., 2005; Singh 64 et al., 2011). Far less is known about past changes in PP in the BoB and their link to changes in monsoon precipitation, although reconstructions and climate model simulations have clearly pointed important changes in 65 ISM precipitation driven by both, orbital forcing and fast changes at high-latitude, such as those associated with 66 67 the Atlantic meridional overturning circulation (AMOC) (e.g. Braconnot et al., 2007a, 2007b; Kagevama et al., 2013; Marzin et al., 2013; Contreras-Rosales et al., 2014). The poorer attention devoted to past PP in the BoB is in 68 part due to the absence of high time-resolution PP records in the BoB and the Andaman Sea (Phillips et al., 2014; 69 Da Silva et al., 2017; Li et al., 2019), which precludes our complete understanding of how monsoon climate changes 70 impact tropical ocean ecology through different mechanisms and at different time-scales. To fill this gap, reliable 71 72 paleo-PP records are needed for that region.

Coccolithophores are marine calcifying phytoplankton organisms that constitute one of the most important "functional groups", responsible for primary production and export of carbonate particles (i.e. the coccoliths they produce) to the sedimentary reservoir. The coccoliths preserved in marine sediment are good study material for paleoenvironmental reconstructions. Particularly, *Florisphaera profunda* is a lower photic zone dweller and its relative abundance in marine coccolithophore assemblages obtained from the sediments has been successfully used to reconstruct past changes of the nutricline depth and PP (Molfino and McIntyre, 1990a, 1990b; Beaufort et al., 1997; Zhang et al., 2016; Hernández-Almeida et al., 2019).

In this study, we provide the first record of coccolith assemblage changes in the BoB. The relative abundance 81 of F. profunda in the sediment core MD77-176 makes it possible to reconstruct at a high-temporal resolution, 82 paleo-PP over the last 26 kyr BP in the northeastern BoB. The studied period covers a complete precession cycle 83 and the last deglaciation. This time interval is characterized by rapid climate changes remotely controlled by north hemisphere high-latitude climate and disruptions of the AMOC (McManus et al., 2004; Clement and Peterson, 84 85 2008; Liu et al., 2009; Wolff et al., 2010; Clark et al., 2012), as observed during the cold periods Heinrich Stadial 1 (17–14.8 kyr BP) and Younger Dryas (12.9–11.8 kyr BP) when massive collapses of northern hemisphere ice 86 shelves release prodigious volume of icebergs and freshwaters in the North Atlantic Ocean (Heinrich, 1988). In 87 88 addition, we used the outputs of paleoclimate experiments obtained with the "Institut Pierre Simon Laplace" Earth 89 System Model version 5 (IPSL-CM5A-LR) (Dufresne et al., 2013) in which marine biogeochemistry is represented, and the transient climate simulations run with the Community Climate System Model version 3 (CCSM3) (He et 90

al., 2008; Collins et al., 2006a), to analyze our reconstructed PP results in terms of local evolution of upper seawater
stratification, as well as monsoon climate dynamics. Based on our reconstructed PP record and modelling results
documented through integrated PP maps as well as oceanic parameters profiles and cross plots, we unravelled the
dynamical relationship between PP in the northeastern BoB and the Indian Monsoon at both orbital- and millennialtimescales.

#### 96 2. Site description and oceanographic setting

Core MD77-176 (14°30'5"N, 93°07'6"E) was retrieved from the northeastern BoB, at the junction with the 98 Andaman Sea, during the OSIRIS 3 cruise of the R.V. *Marion Dufresne* in 1977 (Fig. 1a). The site lies ~200 km 99 southwest of the modern Irrawaddy River mouth. It was retrieved on the continental slope, at a water depth of 1375 100 m, i.e. above the modern lysocline located between ~ 2000 m and ~ 2800 m in the northern BoB (Cullen and Prell, 1984; Belyaeva and Burmistrova, 1985). The lithology consists of olive grey terrigenous clay and silty clay layers 102 with foraminifera- and nannofossil-bearing oozes.

At our core site, the lowest and highest SST recorded during winter ( $\sim 26$  °C) and summer ( $\sim 28-29$  °C), 103 respectively, reflect well the relatively low amplitude SST seasonal changes ( $\sim 2-3^{\circ}$ C) observed in the area 104 (Locarnini et al., 2010). The oceanic environment is under the influence of the Indian Monsoon and shows strong 105 106 seasonal variations in evaporation/precipitation that is expected under such conditions (Webster et al., 1998; Schott and McCreary, 2001; Shankar et al., 2002; Gadgil, 2003). During the summer, moisture-rich southwesterly surface 107 winds blowing from the Indian Ocean result in heavy precipitation (Fig. 1b, d; Lau et al., 2000; Chen et al., 2003; 108 109 Randel and Park, 2006). During winter, dry and cool northeasterly surface winds, weaker than the summer winds, blow from Himalayan highlands and result in drier conditions (Fig. 1c, e). 110

The summer precipitation rates over the BoB and the Andaman Sea and over the surrounding lands (up to 15 mm/day) are much higher than that in the AS (1-3 mm/day; Fig 1d, k). This heavy precipitation area covers the 112 113 catchments of Ganges-Brahmaputra-Meghna (GBM) and Irrawaddy-Salween (IS) river systems, and thus generates massive freshwater discharge (up to 4050 km<sup>3</sup> a year) to the ocean (Sengupta et al., 2006). This input of freshwater 114 depletes sea surface salinity (SSS) at our core site (lower than 33 psu) in the same way as the entire northern BoB 115 116 and Andaman Sea, that is occupied by a low salinity tongue, with a largest extension in November, several months later than the peak of summer precipitation (Akhil et al., 2014; Fournier et al., 2017; Fig. 1f, g, l). Low SSS 117 decreases sea surface density, thereby increasing the density gradient of the upper water column, and thus leading 118 to a strong stratification that impedes the transfer of nutrient from the nutrient-rich deep layer into the euphotic 119 zone. Such a "barrier layer" effect results in generally low annual PP (around 100-140 gC m<sup>-2</sup> yr<sup>-1</sup>) in this area 120

(Prasanna Kumar et al., 2002; Madhupratap et al., 2003; Fig. 1h, i), with maxima being reached during winter, 122 when increased surface wind intensity together with decreased precipitation enhance upper seawater column mixing. The low annual PP at the studied site indicates that this area is not significantly influenced by nutrient 123 124 inputs from rivers to the difference of the near-shore settings, characterized by annual PP maxima (up to 340 gC  $m^{-2} vr^{-1}$ ). By contrast, evaporation is high and precipitation is lowover the AS. It generates higher SSS (higher than 125 35 psu) than in the BoB (Fig. 1f, g, l). Such SSS conditions and therefore, the absence of a strong stratification, 126 makes it possible the development of upwelling and convective mixing during summer and winter respectively. 127 and thus, high PP through the year (up to 320–340 gC m<sup>-2</sup> yr<sup>-1</sup>, Fig. 1h) (Anderson and Prell, 1992; Prasanna Kumar 128 129 et al., 2001, 2009; McCreary et al., 2009).

#### 130 3. Materials and Methods

## 131 3.1 Age model and sampling

The age model of core MD77-176 was previously established by Marzin et al., (2013) based on 31 Accelerator Mass Spectrometry (AMS) <sup>14</sup>C ages combined with the MD77-176 high-resolution oxygen isotope record obtained on planktonic foraminifera *Globigerinoides ruber*, which were correlated to the GISP2 Greenland ice core oxygen isotope curve. The correlated age model is consistent with the AMS <sup>14</sup>C age model, especially after 20 kyr BP (Marzin et al., 2013). The sedimentation rates recorded at site MD77-176 (~25 cm/kyr and up to 40 cm/kyr for the Holocene) provide a good opportunity to study productivity patterns over the last 26 kyr, with millennial to centennial resolutions (Fig. S1).

# 139 3.2 Coccolith analysis and PP reconstruction

For coccolith data, a total of 212 samples were analyzed, with a temporal resolution of ~50 to 400 years. Slides were prepared using the "settling" technique described in Duchamp-Alphonse et al., (2018) after Beaufort et al., (2014). About 0.004 g of dry sediment was diluted in 28 mL Luchon<sup>TM</sup> water (pH = 8, bicarbonate = 78.1 mg/L, total dissolved solid = 83 mg/L) within a flat beaker and settled on a  $12 \times 12$  mm coverslip for 4 h. After pumping the clear liquid out, the coverslip was then dried at 60°C in an oven and mounted on slide with NOA74 glue. This technique ensures a homogenous distribution of coccoliths on the coverslip.

Slices were analyzed with a polarized light microscope (Leica DM6000B) at ×1000 magnification. For each slice, at least 500 coccolith specimens were counted by human eyes under at least 3 random fields of view. The relative abundance of *F. profunda* (Fp%) were calculated as:  $Fp\% = 100 \times (Fp \text{ number / total coccolith number})$ . 149 The 95% confidence interval for Fp% was calculated following the method of Patterson and Fishbein (1984), and 150 corresponds to a reproducibility smaller than  $\pm 5$  %.

Fp% indicates relative depth of nutricline which is critical for PP (Molfino and McIntyre, 1990a, 1990b). To 151 152 the difference of most coccolith species that are found in the upper photic zone (< 100 m water depth), F. profunda 153 dwells in the lower photic zone, at water depth of  $\sim 100-200$  m (Okada and Honjo, 1973). Therefore, when 154 nutricline gets shallower, more nutrient is brought into the upper euphotic zone and primary production increases, while relative abundance of F. profunda decreases. By contrast, when nutricline becomes deeper and primary 155 production decreases, the relative abundance of F. profunda increases. This relationship between Fp% and 156 157 nutricline depth is the basement of PP reconstructions via Fp% in marine sediment. Beaufort et al., (1997) first established a Fp%-PP empirical relationship in the AS based on PP estimates from satellite observations and Fp% 158 in surface sediments. In this study, we estimated PP for the last 26 kyr using a recently published Fp%-PP empirical 159 equation suited for tropical Indian Ocean (Hernández-Almeida et al., 2019):  $PP = [10^{(3.27 - 0.01 \times Fp\%)}] \times 365$ 160 / 1000. The unit of estimated PP is gram of carbon per meter square per vear (gC m<sup>-2</sup> yr<sup>-1</sup>). 161

#### 162 3.3 Paleoclimate simulations

#### 163 3.3.1 Experiments run with IPSL-CM5A-LR

IPSL-CM5A-LR (termed "IPSL-CM5A" in the following) is an Earth System Model (ESM) developed at the 165 "Institut Pierre Simon Laplace" (Dufresne et al., 2013) for the Coupled Model Intercomparison Project phase 5 (CMIP5; Taylor et al., 2012) and the Paleoclimate Modelling Intercomparison Project phase 3 (PMIP3; Braconnot 166 et al., 2012). It is composed of several model components representing the atmospheric general circulation and 167 168 physics (LMDZ5A;Hourdin et al., 2013), the land-surface (ORCHIDEE; Krinner et al., 2005) and the ocean 169 (NEMO v3.2; Madec, 2008) which includes the ocean general circulation and physics (OPA9), sea-ice (LIM-2; Fichefet and Magueda, 1997), and marine biogeochemistry (PISCES; Aumount and Bopp, 2006). The LMDZ 170 atmospheric grid is regular in the horizontal with  $96 \times 95$  points in longitude × latitude (corresponding to a resolution 171 of ~3.75°×1.9°) and 39 irregularly spaced vertical levels. The oceanic grid is curvilinear with 182×149 points, 172 corresponding to a nominal resolution of  $2^{\circ}$ , and 31 vertical levels. It is refined close to the equator, where the 173 resolution reaches  $\sim 0.5^{\circ}$ . 174

Four experiments, set under different boundary conditions, were exploited in this study. Three of them were run for the PMIP3 exercise: the pre-industrial experiment (CTRL), the mid-Holocene experiment (MH), and the Last Glacial Maximum experiment (LGMc). Boundary conditions and details for these three experiments can be found in Le Mézo et al. (2017). The fourth experiment (LGMf) is a freshwater "hosing" simulation similar to the 179 IPSL-CM4 freshwater "hosing" simulation (Kageyama et al., 2013), in which a freshwater flux of 0.2 Sv is applied 180 under LGM (LGMc) conditions over the North Atlantic Ocean, the Nordic Seas, and the Arctic Ocean, that causes 181 the AMOC to slow down (Fig. S2). Both LGMf and LGMc were run for nearly 500 model years. The monthly 182 outputs averaged over the last 100 years of the four experiments were used to compare their mean states. In addition, 183 we focused on monthly results averaged over successive periods of 10 years for the LGMc and LGMf experiments 184 to analyze the transient effects of AMOC changes.

In the glacial experiments (LGMc and LGMf) the sea level is lower, resulting in more extensive continents, including in the study area. The core location is then closer to the coast. In these simulations, the river mouth 186 187 locations, at which fresh water and nutrients from rivers are brought to the ocean, are moved together with the modified coastline. In particular the GBM river mouth is brought to the south of its present location, while the IS 188 river mouth is brought northeastward. These locations have been chosen as they reflect the closest LGM coastal 189 190 points to the present river mouths, and the most probably river paths during low sea-level conditions. In our relatively simple set up, for the MH, LGMc and LGMf simulations we are using the same nutrient content of river 191 192 inputs for the CTRL simulations, in which they are prescribed according to Ludwig et al. (1996). However, due to the sea level drop and associated continental extension under glacial conditions, in LGMc and LGMf, the nutrients 193 194 from rivers are less diluted before reaching a fixed location.

Several parameters were extracted to describe climate conditions: surface wind speed and precipitation minus evaporation rates (P-E), as well as ocean conditions: potential temperature ( $T_{\theta}$ ), salinity, nitrate content (NO<sub>3</sub><sup>-</sup>), upper seawater stratification based on potential density ( $\sigma_T$ ) difference between 200 m and 5 m ( $\Delta$ PD; Behrenfeld et al., 2006), and primary productivity (PP).  $T_{\theta}$  and salinity of the top layer of the oceanic model are used as SST and SSS.

#### 200 3.3.2 TraCE-21 simulation

TraCE-21 (termed "TraCE" in the following) is a transient simulation of the global climate evolution over the last 201 22 kyr which was run with the CCSM3 model designed by the National Center of Atmosphere Research (He et al., 2022008; Collins et al., 2006a; Liu et al., 2009). CCSM3 is a global, coupled ocean-atmosphere-sea ice-land surface 203 climate model, run without flux adjustment (Collins et al., 2006a). It includes four components representing 204 205 atmosphere (CAM3; Collins et al., 2006b), land surface (CLM3; Dickinson et al., 2006), sea ice (CSIM5; Briegleb et al., 2004), and ocean (POP; Smith and Gent, 2002). The forcing of the TraCE-21 simulation comprises changes 206 207in insolation due to the slow variations of astronomical parameters (ORB), in atmospheric greenhouse gases as measured in ice cores (GHG), in topography, land surface type, coastlines (ICE-5G; Peltier, 2004), and in 208freshwater discharge from melting ice sheets which force the AMOC strength to change (MWF; Fig. S3). In 209

addition to the full TraCE-21 simulation, we used the four single-forcing-sensitivity experiments (ORB, GHG, MWF, and ICE), in which only one of the forcing mentioned above is allowed to evolve through time while all the three others are kept fixed at their 19 kyr BP value. Atmosphere decadal-mean seasonal averaged and ocean decadal-mean annual averaged datasets were downloaded from the website of Earth System Grid: https://earthsystemgrid.org/project/trace.html. They have been used to provide the same atmospheric and oceanic parameters simulated by the IPSL model, but over the last 26 kyr, and with the exception of marine biogeochemical variables which are not computed in the CCSM3.

## 217 4. Results

# 218 4.1 Coccolith abundances and reconstructed PP over the last 26 kyr

At the studied site, coccolith assemblages mainly consist of *Florisphaera profunda*, *Emiliania huxleyi*. *Gephyrocapsa* spp. *F. profunda* largely dominates the assemblage (> 60%) over the last 26 kyr, while *E. huxleyi* and *Gephyrocapsa* spp. never exceed 23 % (Fig. 2). Such relative contributions are coherent with coccolith distribution in sediment traps from the northern BoB (Stoll et al., 2007), that shows a high abundance of *F. profunda* due to a strong salinity-related stratification and low surface nutrient concentration (see section 2).

The most striking shifts of coccolith abundances are observed between  $\sim 20$  and  $\sim 11$  kyr BP, and particularly 224 225 around 15-14 kyr BP, when F. profunda drastically increases from 60 to 93 %, while E. huxleyi decreases from 22 226 to 1 % and *Gephyrocapsa* spp. slightly decreases from 12 to 5 %. Such patterns subdivide the record into three main time intervals: (i) from  $\sim 26$  to 19 kyr BP, when F. profunda depicts relatively high amplitude variations, 227 ranging from 60 to 85 % with minima at ~ 25, 23 and 21 kyr BP, while E. huxlevi and Gephyrocapsa spp. both 228 229 average  $\sim 10\%$ ; (ii) from 19 to 11 kyr BP, when F. profunda, E. huxleyi and Gephyrocapsa spp. depicts their highest variations (up to about 33 %, 21 % and 15% in amplitude, respectively) and (iii) from 11 to 1 kyr BP, when F. 230 profunda shows a long-term increasing trend up to 8 kyr, a maxima of 85% between 8 and 6 kyr, and a long-term 231 decreasing trend up to 1 kyr, while *Gephvrocapsa* spp. abundances exceed those of *E. huxlevi* despite minima of 232  $\sim$ 7 % between 8 and 6 kyr BP. 233

Estimated PP varies between 80 and 170 gC m<sup>-2</sup> yr<sup>-1</sup> (Fig. 2). Remarkably, values obtained during the late Holocene (~125 gC m<sup>-2</sup> yr<sup>-1</sup>) are comparable to those recorded in the study area today (annual PP mean of ~135 gC m<sup>-2</sup> yr<sup>-1</sup>). Because estimated PP is inversely related to *F. profunda* percentages (see section 3.2), PP reconstructed over the last 26 kyr mirrors *F. profunda* distribution. It is characterized by peaks higher than 140 gC m<sup>-2</sup> yr<sup>-1</sup> at ~25, 23 and 21 kyr BP. Changes with largest amplitude are found over the deglaciation with a maximum (~170 gC m<sup>-2</sup>  $239 \text{ yr}^{-1}$ ) and a minimum (~80 gC m<sup>-2</sup> yr<sup>-1</sup>) observed at ~15 and 14 kyr BP, respectively. Relatively low PP are recorded 240 during the Holocene, with minima of 90 gC m<sup>-2</sup> yr<sup>-1</sup> obtained between 8 and 6 kyr BP.

#### 241 4.2 Simulated PP and physicochemical profiles

Simulated annual and seasonal (summer and winter) patterns of PP (gC m<sup>-2</sup> vr<sup>-1</sup>) are shown for the BoB and the 242 243 Andaman Sea in Figure 3, where the MH and LGMc simulations are compared to the CTRL one, and where the 244 LGMf simulation is compared to the LGMc one, highlighting the effects of the AMOC slowdown. According to the CTRL simulation, the coastal northern BoB and Andaman Sea as well as the southwestern BoB appear to be 245 246 the most productive areas under pre-industrial conditions, which is in accordance with the Vertical Generalized 247 Production Model (VGPM), representing in situ PP distribution based on satellite derived Chlorophyll concentration (Fig. 1h, i; Behrenfeld and Falkowski, 1997). In all cases, high PP (> 220 gC m<sup>-2</sup> yr<sup>-1</sup>) is related to 248 high nutrient contents in the upper column, thanks to the influence of river discharges (northern coastal BoB and 249 250 Andaman Sea) or the development of coastal upwelling (southwestern BoB; Vinayachandran et al., 2004). Hence, 251 despite its coarse spatial resolution, the IPSL-CM5A model is able to represent the main area of high PP and their seasonal cycles. The differences of annual PP between MH and CTRL reveal a dipole structure in the studied area, 252 with slightly lower PP in the western part of the BoB and slightly higher PP in the eastern part including the 253 Andaman Sea. Strong signal of lower PP is found in the southwestern BoB during summer, and in the northern 254 BoB during winter. Slightly higher PP is found in the eastern BoB and the Andaman Sea, mainly during summer. 255 The overall increase in annual PP simulated within the center part of the BoB during LGM compared to 256 preindustrial (LGMc-CTRL), reflects well the general PP increase simulated during the summer season. This area 257 is an extension of the high PP found by the CTRL simulation within the southwestern BoB. One of the most striking 258 259 pattern highlighted by this comparison is the important increase in annual PP in the northeastern part of the BoB at the junction with the Andaman Sea, that reflects significant increases of PP during both summer and winter seasons, 260 261 while PP in the northern BoB and the whole Andaman Sea is lower. This pattern highlights the CTRL river mouth 262 grids shift toward the northeastern BoB during the LGM (section 3.3.1), and its most probable location closer to the MD77-176 site at that period. Between LMGf and LGMc, PP is lower in the entire BoB, except in the 263 northeastern part of the BoB in winter, for which a higher annual PP is simulated. 264

Summer and winter vertical profiles are extracted from grids at the GBM and the IS river mouths for CTRL and
MH (Fig. 4), and from grids at the northeastern BoB, near the location where core MD77-176 has been retrieved,
for CTRL, MH, LGMc and LGMf (Fig. 5).

CTRL and MH river mouth profiles depict PP maxima within the surface layers (0-50 m), where reduced 270 salinity and density conditions help maintaining a nutricline around 50 m water depth in both seasons (Fig. 4). In 271 all cases, temperatures and SSS are lower during the MH compared to CTRL. The most striking difference is 272 observed for the GBM river mouth system, where salinity is clearly lower within the surface layer (0-30 m) during 273 MH compared to CTRL, especially during winter, while temperature change is limited. Such pattern results in 274 lower density in the surface layer and stronger density gradient within the upper 200 m of the seawater (i.e. stronger salinity-related stratification) during winter season of MH. Under such conditions, nutrient content and thus PP, 275 are lower in the upper 30 m water depth. 276

For the CTRL profiles of the northeastern BoB, PP maxima are found at ~75 m water depth, just above the nutricline, in both seasons (Fig. 5). Such a pattern reflects well what is found in the open sea environment of the BoB at present (Madhupratap et al., 2003). MH PP profiles show no large difference compared to the CTRL ones. It is only during winter, that salinity is significantly lower between 0 and 50 m water depth, and that the associated increase in the density gradient within the photic zone is related to slightly lower PP.

PP profiles of LGMc and LGMf are very different from those of CTRL and MH. They are associated with generally saltier and/or colder surface waters. Interestingly, high PP is found in the surface layers (0–50 m) where 283 284 nutrient contents are higher than CTRL and MH conditions (Fig. 5). Such distributions show that nutrient content 285 and PP are comparable to those found in the CTRL river mouth profiles, and particularly case during winter, where 286 LGMc and LGMf simulations of salinity gradient show a shallower halocline that rises the density gradient of 287 surface layers and is thus accompanied by a shallower pycnocline. It indicates that PP reacts to the shift from the 288 open sea environment configuration during CTRL and MH simulations to the more coastal one during LGMc and 289 LGMf simulations, as previously documented in section 3.3.1. Interestingly, during the winter, PP and nutrients 290 contents between 0 and 30 water depths of LGMf are higher than those of LGMc. Such patterns are associated with 291 higher salinity in surface waters and a reduced density gradient that might promote upper layer mixing. Overall, 292 the LMGc, LMGf and MH simulations do not show strong difference in the vertical variation of temperature 293 compared to the MH. Changes in PP and nutrient contents are rather associated to modifications in density gradient, 294 thanks to salinity changes which highlight the importance of salinity-related stratification vs mixing in the PP 295 distribution in the past.

#### 297 5. Discussion: Forcing factors behind PP variations over the last 26 kyr as revealed by a model-

#### 298 data comparison

#### 299 5.1 The last glacial period

During the LGM (23–19 kyr BP), i.e. when drier conditions prevailed in the area, our reconstructed PP estimates 300 average  $\sim 120$  gC m<sup>-2</sup> yr<sup>-1</sup>, which is nearly the same value as the one reconstructed for the late Holocene (2–1 kyr 301 BP) (Figs. 6i and 7f, g). An important discovery is the high-amplitude millennial-scale variations that PP depicts 302 from 26 to 19 kyr BP. Such variations mirror those of SSS (seawater  $\delta^{18}$ O anomaly signal) obtained on the same 303 core (Fig. 6h), and to some extent in the Andaman Sea (Fig. 6e, g), thus documenting high PP intervals at times of 304 low SSS pulses and vice versa. In such a context, the most plausible explanation for higher PP coeval with low 305 SSS deals with higher nutrient inputs from rivers. Indeed, during the LGM and relatively low sea-level, more 306 307 proximal IS river mouth system might promote freshwater and terrigenous nutrient transfer to our core site, thus decreasing (increasing) SSS and increasing (decreasing) nutrient content and PP, according to South Asia 308 precipitation and riverine flux dynamics. Such millennial-scale variations are readily seen in the record of South 309 Asian monsoonal precipitation, thus confirming our assumption. Indeed, despite long-term aridity during the LMG, 310 as documented by the net precipitation results of the TraCE simulation together with  $\delta^{18}$ O and  $\delta$ D alkane signals 311 from cave speleothems and marine sediments respectively (Fig. 6c, d), rapid SSS decreases at our core site are in 312 phase with short-term increases in precipitation and vice versa (Figs. 6h). They are also found in IPSL-CM5A 313 simulations where higher PP and higher nutrient contents within the upper 50 m of the photic zone during LGMf 314 and LGMc compared to MH and CTRL, reflect higher terrigenous nutrient inputs to the studied site, as the the IS 315 river mouth system migrates probably northward, i.e closer to our core site (section 4.2). Interestingly, the highest 316 reconstructed PP (~160 gC m<sup>-2</sup> yr<sup>-1</sup>) remains lower than the simulated PP at river mouths (>220 gC m<sup>-2</sup> yr<sup>-1</sup>), thus 317 suggesting that core MD77-176 is not within the coastal environment during the LGM, but is rather influenced by 318 the nutrient enriched-river system plume. The local specificities of the area have in part been highlighted by 319 Sijinkumar et al. (2016) that reported lower SSS compared to the modern time in the northern Andaman Sea due 320 to major changes in basin morphologies between both periods, thanks to the sea-level significantly lower during 321 322 the LGM compared to modern times. Therefore, in such contexts, one cannot exclude that both, the low sea-level 323 conditions and the migration of the IS river mouth system, might result in the specific SSS and PP records obtained at our core site. In all cases, it appears that between 26 and 19 kyr BP, the IS river system renders MD77-176 PP 324 325 sensitive to millennial-scale variations in South Asian monsoonal precipitation, as it modulates riverine flux and the extent of the nutrient-rich riverine plume in the area. 326

### 327 5.2 The last deglaciation

During the last deglaciation (19–11 kyr BP), the most striking changes of reconstructed PP covary positively with 328 329 SSS, especially after the 19-17 kyr BP transient period, when high PP intervals correspond to high SSS ones, and 330 vice versa (Fig. 6h, i). Both signals show three stages that correspond to abrupt temperature changes in the North Atlantic Ocean, i.e. the cold Heinrich Stadial 1 (HS1; 17–14.8 kyr BP), the warm Bølling-Allerød (B-A; 14.8–12.9 331 kyr BP) and the cold Younger Drvas (YD; 12.9–11.8 kyr BP), which are characterized by changes in AMOC 332 333 strength (Fig. 6b, h, i; Elliot et al., 2002; McManus et al., 2004). The AMOC is a component in inter-hemispheric transport of heat (e.g. Liu et al., 2009; Buckley and Marshall, 2016) and its changes in intensity, which are related 334 to inter-hemisphere temperature gradient, have a strong influence on tropical Atlantic (Wang et al., 2004; Peterson 335 et al., 2000; Peterson and Huaug, 2006; Swingedouw et al., 2009), and South Asia rainfalls (Overpeck et al., 1996; 336 337 Barber et al., 1999; Fleitmann et al., 2003; Gupta et al., 2003; Murton et al., 2010; Yu et al., 2010; Cai et al., 2012; Marzin et al., 2013). Cold periods in the North Atlantic are associated with relatively weak AMOC and low 338 339 monsoon precipitation, and vice versa. The relationship between South Asian rainfall and AMOC during the last 340 deglaciation has been studied by Marzin et al., (2013), based on several water hosing experiments run with IPSL-CM4 model. They found a strong positive correlation between the AMOC strength and South Asian summer 341 342 precipitation rates and pointed out that temperature anomalies over the tropical Atlantic Ocean are key elements in modulating the tele-connection mechanisms between the AMOC and South Asian rainfall. It has been proposed 343 that a southward shift of the ITCZ was triggered by low tropical Atlantic Ocean temperatures and weakened AMOC 344 345 during HS1 and possibly the YD (Stocker and Johnsen, 2003; Gupta et al., 2003; Goswami et al., 2006; Li et al., 2008; Pausata et al., 2011; McGee et al., 2014; Schneider et al., 2014). Such variations of moisture are simulated 346 here, in the IPSL-CM5A housing simulation (LGMf), that shows weaker summer winds and drier climate over the 347 AS and South Asia when AMOC is weakened compared to the LGMc simulation (Fig. 7k, n). They are also 348 observed in the TraCE simulation over the deglaciation, with millennial-scale variations of net precipitation being 349 350 mainly forced by changes in AMOC strength, and the colder periods (HS1 and YD) being associated with weaker 351 precipitation (Figs. 6d, S4). More importantly, the reconstructed records and TraCE results, together show that weaker net precipitation intervals correspond to higher SSS ones, which indicates that South Asian net precipitation 352 controls the salinity budget in the BoB and Andaman Sea (Figs. 6d, h). Since SSS and PP variations of MD77-176 353 site are highly correlated to upper seawater density gradient (stratification) while SST remains relatively stable 354 355 (Figs. 6f, h, i), it seems reasonable to propose that during the last deglaciation, PP variations are directly driven by 356 precipitation dynamics through changes in upper water column stratification associated to SSS variations (the socalled "barrier layer" effect). An important finding is that millennial-scale variations of MD77-176 PP are anti-357

phased with those in the western and northern AS (Fig. 6j, k), which are indicators of local summer wind strength. We interpret this anti-phased PP patterns by the fact that weaker summer winds (i.e. reduced PP) over AS, by bringing less moisture to South Asia, result in subdued freshwater inputs within the NE-BoB, that weaken stratification and increase PP. To the opposite, stronger summer winds (i.e. higher PP) over AS, reinforce precipitation over South Asia, enhance freshwater inputs within the NE-BOB, and ultimately decrease PP through enhanced stratification.

364 The relationships between ITCZ, southwesterly winds over the AS, South Asian rainfall, SSS, and stratification over the northern BoB and Andaman Sea, are confirmed by IPSL-CM5A. Compared to LGMc, LGMf 365 366 clearly show higher SSS and weaker stratification, especially in the northeastern BoB, under weakened AMOC condition (Fig. 7k-o). The areas with higher PP in the northeastern BoB, that corresponds to the LGMc river mouth 367 grids, match well those with the largest increase of SSS (Figs. 3j and 7m), indicating that salinity-stratification 368 controls PP, even under unchanged amount of nutrient inputs from rivers (section 3.3.1). The relationship that 369 exists between the salinity-stratification and PP of these grids is shown in Fig. 8. It clearly shows a positive 370 371 correlation between PP and nitrate contents and between nitrate contents in the upper photic zone (0-50 water)depth) and SSS. In such a context, PP is therefore inversely correlated to the stratification, with high PP being 372 373 associated to high nutrients, high SSS and reduced vertical density gradient. Moreover, the annual simulated PP increase is mainly associated to PP increase during winter (Fig. 31), which mirrors well the winter peak of PP 374 375 observed in modern times (Fig. 1m).

376 Although LGMf is not set under the complete conditions of HS1 or YD (higher atmospheric pCO<sub>2</sub> and sea-377 level compared to the LGM), it helps deciphering the control that salinity-stratification exerts on PP in the northeastern BoB under weakened AMOC condition and lower South Asian rainfall. Together with the robust 378 relationships that exist between reconstructed PP, SSS, South Asia rainfall, and AS southwesterly winds, we can 379 380 conclude that as the sea-level rises during the last deglaciation, the location of MD77-176 is less influenced by nutrient inputs from the IS river mouth system than during the last glacial period, and that the "barrier layer" effect 381 382 dominates. Therefore, PP variability is highly controlled by the changes of salinity-stratification that is linked to the changes of AMOC strength and monsoon precipitation. 383

## 384 5.3 The Holocene

385 During the Holocene (11–1 kyr), long-term decreasing trends in reconstructed PP match long-term decreasing 386 trends in SSS, increasing trends in South Asian precipitation, and increasing trends in AS PP, while simulated SST 387 show a gradual increase of  $\sim$ 1°C across the area (Fig. 6). Therefore, the relationships between these parameters are 388 similar to those we observed over the last deglaciation. The most obvious pattern is found during the early-middle

Holocene (8–6 kyr BP) when PP and SSS minima correspond to South Asian precipitation and AS PP maxima. 389 390 This time interval, also called the Early Holocene Climatic Optimum (EHCO; e.g. Ciais et al., 1992; Contreras-Rosales et al., 2014), is characterized by higher North Hemisphere (NH) summer insolation compared to present, 391 392 as highlighted by a maximum in the 30°N August mean insolation (Fig. 6a), and the peak of insolation difference 393 between 6 kyr BP and present day over low- and mid-latitude areas (Marzin and Braconnot, 2009). Under enhanced 394 boreal summer insolation, the MH simulation reveals stronger southwesterly summer winds over the AS and enhanced net precipitation over South Asia (Fig. 7p, s), thanks to the northward shift of the ITCZ system (Bassinot 395 et al., 2011; McGee et al., 2014; Schneider et al., 2014). Lower SSS and higher density gradient (stronger 396 397 stratification) are concomitantly documented over the entire BoB, but they are particularly obvious in the northern 398 BoB (Fig. 7q, r), that are directly influenced by freshwater budget and input from the GBM river system (Behara 399 and Vinayachandran, 2016). All these elements suggest that during the Holocene PP changes in the northeastern BoB were most probably driven by salinity-stratification associated to the changes in precipitation. This is 400 401 confirmed by the comparison between the MH and CTRL profiles of the GMB river mouth system, that highlights lower nutrient contents and PP in the upper seawater associated with reduced SSS and increased density gradient 402 403 between 0 and 30 m water depths (section 4.2, Fig. 4).

#### 404 6. Conclusion

We document for the first time, a 26 kyr PP record for the northeastern BoB using an empirical equation relating 405 Fp% to PP. Comparisons of this PP signal with previous geochemical data and new model outputs helped us 406 proposing two coherent scenarios behind PP distribution during 26–19 kyr BP and 19–1 kyr BP intervals, 407 408 respectively. In all cases, PP is related to nutrient content and distribution in sea surface. From 26 to 19 kyr BP, 409 when drier and lower sea-level conditions prevailed, millennial-scale PP changes are most probably related to nutrient discharges from the Irrawaddy-Salween river mouth system, that are paced by South Asian monsoon 410 precipitation changes. Over the last 19 kyr, while the sea-level rise and more humid conditions prevailed, 411 412 millennial-scale PP variations over the deglaciation and long-term trend over the Holovene are rather controlled by salinity-related-stratification that monitor nutrient distribution within the photic zone and is therefore less 413 414 influenced by nutrient inputs from the IS river mouth system. We demonstrate more generally that stratification 415 dynamic during that period, is driven by Indian Monsoon precipitation changes, that generates changes in freshwater supplies to the ocean. The analysis of climate model outputs provides additional evidences for that 416 salinity-stratification hypothesis and help demonstrating that palaeoceanographic changes are forced by AMOC 417 dynamic during the last deglaciation, and insolation during the Holocene. 418

# 419 Data avalibility

420 Coccolith data of core MD77-176 can be found in the supplementary materials.

#### 421 Supplement

422 The supplement related to this article is available online

## 423 Author contribution

XZ, SDA, MK and CC developed the idea. CC and FB provided sediment samples. XZ did coccolith analysis and
visualization of the climate modelling results. The datasets of climate model IPSL-CM5A-LR were provided by
MK. FB and LB joined the discussion and gave additional ideas for the manuscript. All authors contribute to the
manuscript writing.

# 428 Competing interests

The authors declare that they have no conflict of interest.

## 430 Acknowledgements

X. Zhou was supported by a PhD scholarship from the China Scholarship Council (CSC) and thank Laboratoire
des Sciences du Climat et de l'Environnement (LSCE) for admitting his study related to climate model. He also
thanks Hongrui Zhang for personal communication. This research was supported by the French Centre national de
la recherche scientifique (CNRS).

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
