# Peer review of "Dynamics of primary productivity in the northeastern Bay of Bengal over the last 26,000 years"

_Climate of the Past, 2020_

## Referee Comment (RC1) · Anonymous Referee #1 · 10 Apr 2020

General comments:

Zhou et al – present a new primary productivity (PP) record since the 26 ka from the Bay of Bengal and assess the potential drivers of PP variability during the same period. Hereafter, I provide detailed review of the manuscript by highlighting the main strengths and shortcomings of the manuscript. To briefly summarize, the manuscript requires major revision before publication and section-by-section review of the manuscript are presented following general comments.

Merits:

PP variability across timescales in the Bay of Bengal is not as very well researched as it is in other oceanographic settings such as the Arabian Sea, where strong summer

monsoon winds are known to promote upwelling and PP. This study, which presents high-resolution PP proxy data from the Bay of Bengal spanning the last 26 kyrs, is therefore a very welcome contribution. The fact that the study integrates model outputs and proxy data to unravel the relationship between PP variability and monsoon intensity is also commendable. The manuscript addresses relevant scientific questions within the scope of CP and to my knowledge, provides the first record of its kind from the Bay of Bengal. The manuscript meets several of CP peer review guidelines/criteria:

1. Does the paper address relevant scientific questions within the scope of CP? Yes. 2. Does the paper present novel concepts, ideas, tools, or data? Yes. 3. Are the scientific methods and assumptions valid and clearly outlined? Yes. 4. Is the description of experiments and calculations sufficiently complete and precise to allow their reproduction by fellow scientists (traceability of results)? Yes. 5. Do the authors give proper credit to related work and clearly indicate their own new/original contribution? Yes. 6. Does the title clearly reflect the contents of the paper? Yes. 7. Does the abstract provide a concise and complete summary? Yes. 8. Are mathematical formulae, symbols, abbreviations, and units correctly defined and used? Yes. 9. Are the number and quality of references appropriate? Yes (few references are suggested).

Critique:

The overall structure of the manuscript and occasional lack of clarity in some sections is a major shortcoming of the manuscript. For example, results from the model outputs are not fully integrated with proxy data and are rather independently summarized. Although, this manuscript presents an important dataset, which is of interest for the scientific community, some of the interpretations need to be significantly refined and I find few of them not convincing at all (see my comments on the discussion section below). The fact that only figures are provided as the supplementary information is also unhelpful and I believe a short text summary is warranted. To summarize, the manuscript in its present form does not meet the following CP peer review guidelines/criteria:

1. Are substantial conclusions reached? Needs to be improved (see detailed comments below). 2. Are the results sufficient to support the interpretations and conclusions? In general yes but some of the interpretations need to be improved 3. Is the overall presentation well structured and clear? Needs to be improved. 4. Is the language fluent and precise? In general yes but there is occasional lack of clarity in some sections. 5. Should any parts of the paper (text, formulae, figures, tables) be clarified, reduced, combined, or eliminated? Some discussion sections can be combined to improve clarity and train of thought. 6. Is the amount and quality of supplementary material appropriate? The supplementary information lack sufficient information and needs to be significantly improved.

Detailed reviews (section–by–section) 1) Introduction

The introduction section of the manuscript provides sufficient literature review and is quite clear in identifying the key knowledge gaps in the literature. Overall, the introduction section is generally admissible but few descriptions need to be refined. The introduction section establishes originality and the comments provided hereafter are, therefore, mostly typographical, intended to improve overall clarity of the Introduction section.

Line 30: The way the monsoon is currently defined need to be improved.

Here, the monsoon is practically presented as a giant sea breeze that is responsive to changes in the land-sea thermal contrast alone and excludes the more complex aspects of the monsoon and its relation with the tropical ocean on seasonal to interannual to decadal timescales (e.g. ENSO and IOD).

Line 40–53: The subsequent section provides a detailed summary of the oceanographic setting in the Bay of Bengal and Andaman Sea. To be more articulate and improve clarity, it's probably best that comparisons within the Arabian Sea and differences with in the broader Northern Indian Ocean oceanography are presented in the introduction section.

2) Site description and oceanographic setting

This section provides a detailed summary of the oceanographic setting of the studied site and is well written.

Are there any notable differences in seasonal PP variability between the Bay of Bengal and the Andaman Sea? Perhaps a sentence or two addressing the above question will be helpful.

3) Materials and Methods

This section provides a detailed summary of the methodology and is generally well written. However, information provided on age model reconstruction is insufficient and citation of Figure 2 is not very useful either. I suggest that the authors provide a summary of the age model including changes in the rate of sedimentation etc. This can be included in the same section or in the form of a supplementary material.

4) Results and discussions

This section of the manuscript is poorly structured and in my opinion, the weakest part of the manuscript. For example, a large chunk of the text (e.g. section 4.3, section 4.3.1: lines 300 – 317) should have been included in the methodology section. This has made the discussion section overall very descriptive and lacking in substance, and most crucially hard to follow. One way of overcoming this predicament is to divide this section in to two separate sections (i.e., Results and Discussions). For example, the proxy data and model data results can be grouped into two subsections and the discussion section should focus on the dynamics of PP variability over the studied time interval. The Discussion section should also integrate both proxy data and model inferences to build a more coherent understanding of PP variability over the last 26 kyrs.

Looking at the PP record, it is clear that there are three distinct time intervals that can be discussed separately including the highly variable LGM (?), the last deglaciation

period marked by an abrupt shift in PP centered around the BA and the Holocene period, which displays a more gradual change. Therefore, dividing the discussion section accordingly and zooming on these three distinct periods will significantly improve clarity.

Lines 205 – 208: the authors write 'at millennial-scale, large magnitude PP oscillations, are observed during the deglaciation (19–11 kyr BP), showing similar features than those found in the Greenland ice core $\delta$18O record, representing the rapid climatic changes in north hemispheric high-latitude areas (Fig. 2; Stuiver and Grootes, 2000).'

But it is stated in section 3.1 that the age model, although primarily based on 31 AMS 14C dates, it was still tuned to GISP2 Greenland ice core $\delta$18O curve. Can this be considered circular reasoning?

Lines 255 – 229: the authors write, 'Several pieces of evidence suggest that millennial-scale variations of PP between 26 and 19 kyr (i.e. before the LGM) chiefly resulted from wind-driven mixing. First, high PP values are reached during intervals of low surface water salinity. If these PP variations (and upper water column stratification) were primarily driven by precipitation – evaporation changes, the opposite relationship would be expected, and PP would peak at periods of higher salinity because of the weaker barrier layer effect'.

Can you independently verify if the wind-driven mixing in the Northern Indian Ocean was enhanced during the LGM? Which are the intervals of low salinity during the LGM? Isn't the LGM Andaman Sea significantly more saline compared to other periods such as the Holocene? How does precipitation minus evaporation impact PP variability in general? What inferences can be made on LGM PP variability from the LGM experiments? These questions, hopefully, will trigger more in-depth discussions on the LGM oceanography and monsoon paleoclimatology and a thorough analysis of the dataset at your disposal (both proxy observations and model outputs).

Lines 247– 251: the authors write, 'The factors controlling PP on the millennial-scale

over the last 19 kyr, appear to differ from those acting before the LGM. A strong impact of wind-related changes on vertical stratification is unlikely given that river runoff (Gebregiorgis et al., 2016), together with the precipitation (Dutt et al., 2015; Contreras-Rosales et al., 2014), gradually strengthened in this area during the last deglaciation up to the mid-Holocene, due to stronger southwest monsoon circulation (Fig. 4c–g)'.

Although, this is interpretation is well supported, it should also be noted that the PP record during the last deglaciation is also marked by a more abrupt transition. One of the explanations provided involved the collapse of the AMOC and given that stratification changes in the Bay of Bengal/Andaman Sea are driven by changes in monsoon intensity, how does the AMOC (oceanographic processes) impact the monsoon (predominantly atmospheric in nature)? Is the position of the ITCZ the only viable explanation? Note that important clues that will help answer these questions are provided in lines 294 – 299 and section 4.3.2. This is why I recommended complete restructuring of the discussion section.

In section 4.1 (line 217): The authors write that, 'PP peaks are related to low SSS intervals before the LGM, and high SSS intervals over the last 19 kyr'.

Although, PP did not significantly change over the course of the Holocene, there appears to be a clear discrepancy between the gradual monsoon intensification over the Holocene and PP variability. PP variability over the course of the last deglaciation and the Holocene are clearly different. Proxy data shown in Figure 2 suggest that estimated PP has lower valued during the Mid-Holocene ($\sim$90 gC m-2 yr-1) compared to late – Holocene ($\sim$130 gC m-2 yr-1). This, however, is not discussed in any detail and the way the discussion section is structured is at fault again.

In section 4.1 (lines 204 – 205): it is briefly mentioned that PP variability shows 'an opposite trend compared to insolation (Fig. 4a, h)' and in section 4.3.1 it is stated that 'insolation is the main climate forcing factor during the Holocene'.

Why do we have the monsoon peaking later during the mid-Holocene lagging maximum

[Figure]

Northern Hemisphere summer insolation by few kyrs then? The lagged response of the monsoon to insolation forcing suggests that orbital scale monsoon variability is more complex (see Clemens et al., 2003; Caley et al., 2011; Gebregiorgis et al., 2018). Having this in mind, I would therefore encourage the authors to have a more critical outlook on PP variability over the course of the Holocene. I also recommend including Figure S3 – S6 in the main text body and can be used to gain some unique insights on LGM, deglacial and Holocene PP variability. Perhaps Fig. 3 can be moved to the supplementary section.

5) Conclusion

Provides a good summary of the main findings of the paper in its present state and I look forward to reading the revised version of the manuscript. Well done!

Additional comments for the authors

Line 57: What 'fast changes'? Please rephrase.

Line 61: PP record or paleo-PP record. Stick with one for consistency.

Line 61: Da Silva et al., 2017 is a relevant reference here.

Line 62: 'tropical ocean ecology' is very broad and I am not sure this is accurate as well. Perhaps Northern Indian Ocean ecology is more appropriate.

Line 73: 'High-time-resolution' or 'High-temporal-resolution'? 'High-resolution' is a perhaps a better phrase.

Line 74: Why is it important that the 'studied period covers a complete precession cycle'? This sentences need to be qualified or delete otherwise.

Line 80: 'interpret' is perhaps a better word here than 'analyze'.

Line 199–200: 'At orbital scale' – remove.

Line 202: use 'maximum or minimum Northern Hemisphere (NH) summer insolation'

instead of low or high insolation with no reference to the latitude or the season.

Line 205: 'On millennial timescale...'

Line 211: 'Synchronous vs. asynchronous' rather than 'Negatively vs. positively correlated' and of course 'correlation' being a statistical term.

Line 290–291: Rephrase or remove

'During the Holocene, insolation is the main climate forcing factor since other forcing (i.e. greenhouse gas, ice volume, coastlines, vegetation) are relatively stable after the deglaciation.

Line 291–292: Rephrase. Perhaps, a sentence along these lines will do: 'the response of the Indian monsoon to changes in orbital insolation has previously been examined using both AGCMs and ocean–atmosphere general circulation models. . .(Refs).)'

'The mechanisms that force monsoon climate to change were studied by many modeling works (Refs).'

Figures S1, S5, S6 are not cited in the main text and please add supplementary text to the supplementary information. Also make sure that the figures are in chronological order.

References

Caley, T., Malaizé, B., Zaragosi, S., Rossignol, L., Bourget, J., Eynaud, F., Martinez, P., Giraudeau, J., Charlier, K. and Ellouz-Zimmermann, N., 2011. New Arabian Sea records help decipher orbital timing of Indo-Asian monsoon. Earth and Planetary Science Letters, 308(3-4), pp.433-444.

Clemens, S.C. and Prell, W.L., 2003. A 350,000 year summer-monsoon multi-proxy stack from the Owen Ridge, Northern Arabian Sea. Marine Geology, 201(1-3), pp.35-51.

Da Silva, R., Mazumdar, A., Mapder, T., Peketi, A., Joshi, R.K., Shaji, A., Mahalakshmi, P., Sawant, B., Naik, B.G., Carvalho, M.A. and Molletti, S.K., 2017. Salinity stratification controlled productivity variation over 300 ky in the Bay of Bengal. Scientific reports, 7(1), pp.1-7.

Gebregiorgis, D., Hathorne, E.C., Giosan, L., Clemens, S., Nürnberg, D. and Frank, M., 2018. Southern Hemisphere forcing of South Asian monsoon precipitation over the past∼ 1 million years. Nature communications, 9(1), pp.1-8.

---

## Referee Comment (RC2) · Anonymous Referee #2 · 22 Apr 2020

The manuscript by Zhou and co-authors is interesting, adding new insights in the monsoon-PP connections in a region which is relatively understudied compared to other parts of the Indian Ocean. The combination of proxy and model outputs helps to understand better the relationship between salinity and stratification in the water column, beyond the reasonable assumptions anyone can made about the link between salinity and productivity. I think the overall quality of the manuscript is good, and deserves publication in Climate of the Past after some moderate revisions.

The structure of the manuscript needs some improvement. While the first part (Introduction, Material and Methods) are very well written (although lacking some details about age-model), the discussion relative to the model outputs is not so clear. I find the discussion about model output section very difficult to follow, needs to be simplified

in order to improve the understanding and to be better integrated with the proxy data, not to be discussed separately. The figure 2 is not very useful, it repeats data that are shown later in other figures several times. For example, showing the d18Osw and the GISP2 ice-core d18O is not really relevant, as we see the proxy data already tuned to the ice-core data. I assume that Marzin et al., (2013) contains a plot showing this, so these two curves are not needed here. An important point regarding the age-model is that if, despite the large number of radiocarbon ages, the proxy data is tuned to the GISP ice-core d18O, later comparisons between proxy and ice-core data are not very well sustained (circularity). The authors should keep this in mind when discussing about it at L. 205-207.

I find particularly intriguing the change in the salinity-PP relationship before and after LGM (L. 213-222).. The authors suggest that the higher PP during low salinity between 26-19ka are due to higher wind mixing. Are there independent proxy evidence of this coupling? For example, loess deposits that could record changes in wind intensity which could support their view? And why the wind-forcing gets weaker after the LGM?

Finally, in the section Data availability the authors indicate that "Data to this paper can be required. Please contact the X. Zhou or S. Duchamp-Alphonse.". Copernicus journals (including Climate of the Past) have a very clear policy regarding data curation (https://www.climate-of-the-past.net/about/data_policy.html), which "requests depositing data that correspond to journal articles in reliable (public) data repositories, assigning digital object identifiers, and properly citing data sets as individual contributions". Clearly the current statement about data availability does not meet this criteria, and all data and code should be archive somewhere or included as supplementary material.

Some minor corrections: L. 104. Abbreviate Arabian Sea L. 177. Strange symbol between longitude and latitude. Fig. 1f, why choosing SON instead of JJA as the other panels?
* * *

---

## Author Comment (AC2) · 14 Jun 2020

**Response to Reviewer #2**

**Dear reviewer,**

Please find below our answers to the constructive remarks you raised regarding our manuscript below. They all have been carefully considered and will provide what we feel is a much improved manuscript. You will also find, below, all of the modified figures of the new manuscript and Supplementary Material, below.

**Comment #1 (C1):** The structure of the manuscript needs some improvement. While the first part (Introduction, Material and Methods) are very well written (although lacking some details about agemodel), the discussion relative to the model outputs is not so clear. I find the discussion about model output section very difficult to follow, needs to be simplified in order to improve the understanding and to be better integrated with the proxy data, not to be discussed separately.

**Reply #1 (R1):** We agree with Reviewer #2. Therefore, we revised the structure of our manuscript, based (as well) on Reviewer #1' suggestions. An entire chapter is now devoted to the results. In the discussion, empirical data and model outputs are interpreted simultaneously in chapter 5. Particularly, sections 5.1 to 5.3 discuss PP patterns regarding the LGM, the deglaciation, and the Holocene, respectively. We believe that this new structure is helpful to build a more coherent scheme behind PP variability.

Below is the new structure of chapters 4 and 5:

- 4. Results
- 4.1. Coccolith abundances and reconstructed primary productivity over the last 26 kyrs
- 4.2. Simulated primary productivity and physicochemical profiles in the northeastern Bay of Bengal
- 5. Forcing factors behind PP variations over the last 26 kyrs: the inputs of model-data comparisons
- 5.1. During the glacial period
- 5.2. During the last deglaciation
- 5.3. During the Holocene

In detail:

- In section 4.1, we present and describe coccolith species abundances and reconstructed PP (Figure 1 of Author Reponse (Fig. AC1)).
- Section 4.2. relies on new IPSL-CM5A-LR figures dedicated to model results, that help understanding and improving model output interpretations i.e. i) simulated PP maps (Fig. AC2), and ii) simulated vertical profiles of potential temperature, salinity, potential density, and nitrate content of the northeastern Bay of Bengal under four experimental runs (Fig. AC3), that help discussing climate conditions for the LGM (LGMc), the Heinrich Stadial 1 (LGMf), and the Mid-Holocene (MH), compared to preindustrial (CTRL). We show the results of annual mean, summer seasons mean (from June to August, JJA) and winter seasons mean (from December to February) for all these specific time intervals, in order to evaluate PP changes during the monsoonal seasons.

In sections 5.1 to 5.3, we compare our reconstructed PP signal with the published empirical records previously documented in Fig. AC4, and with TraCE-21 transient simulations of the upper water column stratification, SSS, SST and net precipitation (P-E), previously documented in Fig. 5. Merging our previous Figures 4 and 5 into a new figure (Fig. AC4), allows to better discuss PP variations in the monsoonal context. We also combine atmospheric and oceanic outputs of the four experiments run together with the simulated PP obtained by the IPSL-CM5A-LR model in order to better discuss and interpret our reconstructed PP during the last glacial period (section 5.1; Fig, AC5, AC6), the last deglaciation (section 5.2; Fig. AC7, AC8) and the Holocene (section 5.3; Fig, AC9).

**C2:** The figure 2 is not very useful, it repeats data that are shown later in other figures several times. For example, showing the d180sw and the GISP2 ice-core d180 is not really relevant, as we see the proxy data already tuned to the ice-core data. I assume that Marzin et al., (2013) contains a plot showing this, so these two curves are not needed here. An important point regarding the age-model is that if, despite the large number of radiocarbon ages, the proxy data is tuned to the GISP ice-core d180, later comparisons between proxy and ice-core data are not very well sustained (circularity). The authors should keep this in mind when discussing about it at L. 205-207.

**R2:** The initial Figure 2 does not exist anymore. GISP  $\delta^{18}$ O and  $\delta^{18}$ O *G.ruber* obtained on core MD77-169 are now only evoked when dealing with the age model (Figure S1). *Florisphaera profunda* distribution and PP reconstructions are presented within the a new figure (Fig. AC1), that is entirely devoted to micropalaeontological results (i.e. abundances of *F. profunda, Gephyrocapsa* spp. and *Emiliania huxleyi* together with PP estimates).

We thank Reviewer # 2 for highlighting that our phrasing in lines 205-207 could be seen as a circular reasoning, since proxy data are in part tuned to the GISP  $\delta^{18}$ O signal. However, our micropalaeontological data are well in phase with numerous geochemical data obtained elsewhere in the Tropical Indian Ocean and the Chinese continent, based on sediment cores and speleothems with totally independent age models. They also match very well the TraCE 21 and IPSL-CM5A-LR outputs obtained here. Such feature, together with its use in previous works (Marzin et al., 2013; Yu et al., 2017; Ma et al., 2019), point to a robust age model and demonstrate that our micropalaeontological data can be discussed properly in the light of the rapid climatic changes recorded in northern highlatitudes. To avoid any confusion, we rephrased lines 205-207 of the manuscript focusing on the relationship that exists between PP and SSS.

**C3:** I find particularly intriguing the change in the salinity-PP relationship before and after LGM (L. 213-222).. The authors suggest that the higher PP during low salinity between 26-19ka are due to higher wind mixing. Are there independent proxy evidence of this coupling? For example, loess deposits that could record changes in wind intensity which could support their view? And why the wind-forcing gets weaker after the LGM?

**R3:** To our knowledge, there is a high-resolution record of loess grain size from the northeastern China which indicates the local winter wind intensity (Sun et al., 2012; Zhang et al., 2016). The record shows that the winter wind is stronger during LGM than during the late Holocene. However, there is no

published record of wind intensity for the Bay Bengal and Andaman Sea. We think it might be questionable to use the wind record over the northwestern China to interpret the Bay of Bengal as these two regions are not close to one another and the wind directions are different (Fig. 1c; Sun et al., 2012). We checked the modeling outputs and found that compared to preindustrial (CTRL), stronger summer winds and weaker winter winds prevailed over annually saltier sea surface in the Bay of Bengal during the LGM (Fig. AC5, AC6). This implies that the winter wind over the northwestern China and the Bay of Bengal are not strengthened at the same period during the LGM. Therefore, we think if wind-mixing is stronger over the Bay of Bengal during the LGM, it should be related to strengthened summer winds. However, the relationship between PP and SSS of MD77-176 encourages us to explore further mechanisms behind PP (and SSS) variability at that time. We have also found that IPSL-CM5A-LR outputs show spatial differences of SSS in the Bay of Bengal, and particularly, that the studied area could have been associated to low SSS during the LGM (Sijinkumar et al. (2016). Our best explanation is that during the LGM, i.e. under relatively low sea-level, and a more proximal environment for MD77-176, PP and SSS react to the Irrawaddy dynamic in the same way as proximal environment do, today (Fig. 1). Indeed, higher (lower) nutrient and freshwater inputs from the Irrawaddy river, may trigger higher PP and lower SSS, and vice versa. Such assumption is confirmed in Fig. AC2 and AC3 where PP strongly increases (Fig. AC2), when vertical profiles clearly depict a change from open ocean type to coastal one (Fig. AC3). Our scenario seems therefore to provide a suitable explanation behind the PP pattern reconstructed for the LGM.

Bearing in mind that the LGMc experiment of IPSL-CM5A-LR gives us a mean state of PP and SSS conditions and may not simulate the high-resolution PP changes discussed here, we only evoke the possible Irrawaddy river influence on PP distribution during the LGM, with caution.

**C4:** Finally, in the section Data availability the authors indicate that "Data to this paper can be required. Please contact the X. Zhou or S. Duchamp-Alphonse.". Copernicus journals (including Climate of the Past) have a very clear policy regarding data curation (https://www.climate-of-the-past.net/about/data\_policy.html), which "requests depositing data that correspond to journal articles in reliable (public) data repositories, assigning digital object identifiers, and properly citing data sets as individual contributions". Clearly the current statement about data availability does not meet this criteria, and all data and code should be archive somewhere or included as supplementary material.

R4: Thanks for this reminding. We have added our data in the supplementary materials.

*C5:* Some minor corrections: L. 104. Abbreviate Arabian Sea L. 177. Strange symbol between longitude and latitude.

**R5:** We have corrected them.

**C6:** Fig. 1f, why choosing SON instead of JJA as the other panels?**

**R6:** Because the occupation of the input fresh water is the largest during SON in the northeastern Indian Ocean at modern time, lagging the maximum precipitation over the South Asia.

**R7: modified supplementary figures are Fig. AC10 to AC15**

**References**

**Marzin et al., 2013.** Glacial fluctuations of the Indian monsoon and their relationship with North Atlantic climate: new data and modeling experiments, Climate of the Past, 9, 2135–2151.

Yu et al., 2018. Antarctic Intermediate Water penetration into the Northern Indian Ocean during the last deglaciation. Earth and Planetary Science Letters, 500, 67–75.

**Ma et al., 2019.** Changes in Intermediate Circulation in the Bay of Bengal since the Last Glacial Maximum as inferred from benthic foraminifera assemblages and geochemical proxies. Geochemistry, Geophysics, Geosystems, 20, 1592-1608.

**Sijinkumar et al., 2016.**  $\delta$ 18O and salinity variability from the Last Glacial Maximum to Recent un the Bay of Bengal and Andaman Sea. Quaternary Science Reviews, 135, 79–91.

Sun et al., 2012. Influence of Atlantic meridional overturning circulation on the East Asian winter monsoon. Nature Geoscience, 5, 46-49.

**Zhang et al., 2016.** Dynamics of primary productivity in the northern South China Sea over the past 24,000 years. Geochemistry, Geophysics, Geosystems, 17, 4878-4891.

**Figures**

Fig. AC1. Relative abundance changes of main coccolith species and reconstructed PP.

---

## Author Response (AR1)

Xinquan Zhou PhD Student UMR8148 GEOPS xinquan.zhou@universite-paris-saclay.fr Paris Saclay University 91405 ORSAY Cedex, France

Dear Dr. Yin,

Please, find attached our revised manuscript entitled "Dynamics of primary productivity in the northeastern Bay of Bengal over the last 26,000 years" by Zhou et al., which I would like to resubmit as an Article to Climate of the Past. We truly thank the 2 reviewers for their constructive remarks. Their expertise in the field gave us a chance to take some aspects of our work a step further and provide, what we feel is a much improved manuscript. Particularly, we followed the proposition made by Reviewer #1 regarding the structure of our manuscript and as previously explained in our reply to reviewers, an entire chapter is now devoted to the results that are now discussed regarding the three time-intervals highlighted by Reviewer #1 (the last glacial period, the last deglaciation and the Holocene). Besides, model outputs are now fully integrated with proxy data which help better setting our arguments up and present our findings in a clearer way.

In the current revised manuscript, all the fundamental changes mentioned in our previous responses to reviewers have been carefully added, so that the message remains unchanged. We have however merged some figures and removed a few supplementary ones to provide what we feel is a much clearer manuscript. You will find below our previous response to the reviewers with changes marked.

Based on the positive feedbacks we received from the referees, highlighting the novelty of our study, we are further confident that our manuscript, will be of interest to the broad climate science community and will provide a novel insight into a yet overlooked aspect of the relationship that exists between the Indian Monsoon and Primary Productivity in the past.

Our resubmission consists of a central text, eight colour figures and five supplementary figures.

On behalf of the co-authors, Yours sincerely, Xinquan Zhou

**Response to Reviewer #1**

**Dear reviewer,**

Please find below our answers to the constructive remarks you raised regarding our manuscript. They all have been carefully considered and will provide what we feel is a much improved manuscript. You will also find all of the modified figures of the new manuscript and Supplementary Material.

**Comment #1 (C1):** The overall structure of the manuscript and occasional lack of clarity in some sections is a major shortcoming of the manuscript. For example, results from the model outputs are not fully integrated with proxy data and are rather independently summarized. Although, this manuscript presents an important dataset, which is of interest for the scientific community, some of the interpretations need to be significantly refined and I find few of them not convincing at all (see my comments on the discussion section below). The fact that only figures are provided as the supplementary information is also unhelpful and I believe a short text summary is warranted. To summarize, the manuscript in its present form does not meet the following CP peer review guidelines/criteria:

1. Are substantial conclusions reached? Needs to be improved (see detailed comments below). 2. Are the results sufficient to support the interpretations and conclusions? In general yes but some of the interpretations need to be improved. 3. Is the overall presentation well structured and clear? Needs to be improved. 4. Is the language fluent and precise? In general yes but there is occasional lack of clarity in some sections. 5. Should any parts of the paper (text, formulae, figures, tables) be clarified, reduced, combined, or eliminated? Some discussion sections can be combined to improve clarity and train of thought. 6. Is the amount and quality of supplementary material appropriate? The supplementary information lack sufficient information and needs to be significantly improved.

**Reply #1 (R1):** We agree with Reviewer #1 that despite the important dataset we provide in the manuscript, some changes in the structure of the manuscript might help describing our results more clearly, and improving our interpretations and conclusions. Now we clearly separate the result and discussion sections (new sections 4 and 5, respectively), and fully discuss the model outputs together with the empirical data in subsections 5.1 to 5.3 (see our Reply 6). This change relies on the relocation of figures from the Supplementary Material to the main core of the manuscript, and on the addition of new figures related to simulated Primary productivity (PP) and oceanic profiles in the results. The number of figures in the supplementary Material is thus reduced, and we significantly improved the explaining text for all of the remaining figures.

**C2:** Line 30: The way the monsoon is currently defined need to be improved. Here, the monsoon is practically presented as a giant sea breeze that is responsive to changes in the land-sea thermal contrast alone and excludes the more complex aspects of the monsoon and its relation with the tropical ocean on seasonal to interannual to decadal timescales (e.g. ENSO and IOD).

**R2:** In addition to the simple 'sea-breeze' description of the monsoon, there is, indeed, a description that focuses on its energetic aspects and provides a broader overview of the mechanisms behind monsoon variability (Schneider et al., 2014). In the revised manuscript, we now mention both aspects,

and added a few sentences to mention the interannual and decadal changes of monsoon related to ENSO and IOD variability, an important aspect of Indian Monsoon natural variability. It also echoes the seasonal and interannual PP changes we describe in the introduction. However, since the present manuscript chiefly deals with orbital to millennial climate changes, we chose to not fully detail this aspect in the introduction.

**C3:** Line 40–53: The subsequent section provides a detailed summary of the oceanographic setting in the Bay of Bengal and Andaman Sea. To be more articulate and improve clarity, it's probably best that comparisons within the Arabian Sea and differences with in the broader Northern Indian Ocean oceanography are presented in the introduction section.

**R3:** We totally agree with Reviewer #1. Indeed, in this section, we highlight the specific patterns of PP in the Bay of Bengal and the Andaman Sea compared to the Arabian Sea. PP in the Arabian Sea is particularly high compared to the Bay of Bengal and the Andaman Sea during Summer Monsoon, due to the occurrence of important coastal upwelling that bring nutrients into the photic zone. To the contrary, summer monsoon is associated with important freshwater inputs in the Bay of Bengal that cause salinity-driven, water column stratification, resulting in a reduced nutrient input to the upper water column, and thus subdued PP. Such broad PP difference is an important aspect that we also highlight when discussing about past evolution (new section 5) and compare our results with previous works (Schulz et al., 1998; Ivanochko et al., 2005). It seems therefore very important to mention such modern pattern in the introduction.

Lines 40-53 might not be clear enough, particularly when dealing with acronyms such as the Andaman Sea or Arabian Sea. Since we don't refer to the Andaman Sea very often in the manuscript, we only use diminutives for the Bay of Bengal and the Arabian Sea. We also describe more clearly the relationship that actually exists between the upwelling system and PP in the Arabian Sea, adding a few sentences and references (Bartolacci and Luther, 1999 Anderson and Prell, 1992; Madhupratap et al., 1996; Gardner et al., 1999; Wiggert et al., 2005; Liao et al., 2016) on this aspect. We are aware that in the western Arabian Sea, the summer upwelling system is quite complex, with for example, a branch that can transport nutrient to the central part of the Arabian Sea. However, we prefer to not mention PP distribution in a very detailed way, because we are not able to discuss its evolution and distribution with such details in the past due to a lack of high-resolution PP.

**C4: 2) Site description and oceanographic setting**

**This section provides a detailed summary of the oceanographic setting of the studied site and is well written.**

Are there any notable differences in seasonal PP variability between the Bay of Bengal and the Andaman Sea? Perhaps a sentence or two addressing the above question will be helpful.

**R4:** Geographically and oceanographically speaking, our site is located at the junction between the northeastern Bay of Bengal and the northern Andaman Sea. These two parts represent open oceanic settings and are both influenced today by low SSS seawaters originating from the Irrawaddy-Salween river system (Figs. 1g, f). They are both characterized by annual rates of PP around 100-140 gC m-2 yr-1 (Fig. 1h, i). Very high annual PP (up to 340 gC m-2 yr-1) can be observed in coastal settings that

are under the direct influence of river-driven nutrients, but these nutrients are actually consumed in these proximal environments and do not reach the studied site. Such configuration may have changed in the past particularly during the LGM when sea-level was relatively low (see Reply 8). However, there is no reason why the northeastern Bay of Bengal and the northern Andaman sea should behave in a completely different way under such conditions (Fig. 1), and the most likely forcing factor that might drive orbital and millennial PP changes is monsoon, modulated by sea-level, insolation and/or AMOC dynamics. Our core location is therefore suitable to test the relationships between these parameters. As suggested by Reviewer #1, we added a sentence to highlight such similarities between the northeastern Bay of Bengal and the northern Andaman Sea, in the new version of the manuscript.

**C5:** 3) Materials and Methods**

This section provides a detailed summary of the methodology and is generally well written. However, information provided on age model reconstruction is insufficient and citation of Figure 2 is not very useful either. I suggest that the authors provide a summary of the age model including changes in the rate of sedimentation etc. This can be included in the same section or in the form of a supplementary material.

**R5:** The age model used herein has originally been described in Marzin et al., (2013), and has latterly been used by Yu et al., (2018) and Ma et al., (2019). Indeed, Marzin et al. (2013) devoted an entire chapter to this chronological aspect (in their chapter 2.1), and already described all of the important information required herein, such as the sedimentation rate (represented in their Fig. 3). Therefore, we decided to refer to Marzin et al. (2013) but we added a figure including the sedimentation rates of the core within the Supplementary Material (new Figure S1). We discuss this part with extreme caution to avoid any confusions regarding the age model, and clearly demonstrate its robustness. The Figure 2 has been modified compared to the initial submission. It is now new Figure 23 that includes relative abundance of coccoliths and reconstructed PP.

**C6:** 4) Results and discussions**

This section of the manuscript is poorly structured and in my opinion, the weakest part of the manuscript. For example, a large chunk of the text (e.g. section 4.3, section 4.3.1: lines 300 - 317) should have been included in the methodology section. This has made the discussion section overall very descriptive and lacking in substance, and most crucially hard to follow. One way of overcoming this predicament is to divide this section in to two separate sections (i.e., Results and Discussions). For example, the proxy data and model data results can be grouped into two subsections and the discussion section should focus on the dynamics of PP variability over the studied time interval. The Discussion section should also integrate both proxy data and model inferences to build a more coherent understanding of PP variability over the last 26 kyrs.

Looking at the PP record, it is clear that there are three distinct time intervals that can be discussed separately including the highly variable LGM (?), the last deglaciation period marked by an abrupt shift in PP centered around the BA and the Holocene period, which displays a more gradual change. Therefore, dividing the discussion section accordingly and zooming on these three distinct periods will significantly improve clarity **R6:** We believe that the proposition made by Reviewer #1 regarding the structure of our manuscript will certainly clarify it, therefore helping to improve the description of our results as well as the interpretations. Therefore, we changed our manuscript in the light of the suggestions. An entire chapter is now devoted to the results. In the discussion, empirical data and model outputs are interpreted simultaneously, which is helpful to build a more coherent scheme behind PP variability. Our results are now discussed regarding the three time-intervals highlighted by Reviewer #1.

Below is the new structure of chapters 4 and 5:

4. Results

4.1. Coccolith abundances and reconstructed primary productivity over the last 26 kyrs

4.2. Simulated primary productivity and physicochemical profiles in the northeastern Bay of Bengal
 5. Forcing factors behind PP variations over the last 26 kyrs: the inputs of model-data comparisons
 comparisons
 Discussion: Forcing factors behind PP variations over the last 26 kyr as revealed by a model-data comparison

5.1. During tThe glacial period

- 5.2.  $\frac{\text{During t}}{\text{During t}}$  he last deglaciation
- 5.3. During tThe Holocene

In detail:

- In section 4.1, we present and describe coccolith species abundances and reconstructed PP (Figure 1 of Author Response (Fig. AC1)new Fig. 2).
- Section 4.2. relies on new IPSL-CM5A-LR figures dedicated to model results, that help understanding and improving model output interpretations i.e. i) simulated PP maps (Fig. AC2new Fig. 3), and ii) simulated vertical profiles of potential temperature, salinity, potential density, and nitrate content of the Ganges-Brahmanputra-Meghna and Irrawaddy-Salween river mouth systems (new Figure 4) and of the northeastern Bay of Bengal under four experimental runs (Fig. AC3new Fig. 5), under two and four experimental runs, respectively. Itthat helps discussing climate conditions for the LGM (LGMc), the Heinrich Stadial 1 (LGMf), and the Mid-Holocene (MH), compared to preindustrial (CTRL). We show the results of annual mean, summer seasons mean (from June to August, JJA) and winter seasons mean (from December to February) for all these specific time intervals, in order to evaluate PP changes during the monsoonal seasons.
- In sections 5.1 to 5.3., we compare our reconstructed PP signal with the published empirical records previously documented in Fig. AC4, and with TraCE-21 transient simulations of the upper water column stratification, SSS, SST and net precipitation (P-E), previously documented in Fig. 5(new Fig. 6). We have mMergeding our previous Figures 4 and 5 into thea new Figure 6 (Fig. AC4), allows to better discuss PP variations in the monsoonal context. We also combine atmospheric and oceanic outputs of the four experiments run together with the simulated PP obtained by the IPSL-CM5A-LR model (new Fig. 7) in order to better discuss and interpret our reconstructed PP during the last glacial period (section 5.1; Fig, AC5, AC6), the last deglaciation (section 5.2; Fig. AC7, AC8) and the Holocene (section 5.3; Fig, AC9), as proposed by Reviewer #1. A new Figure 8 merged from our previous Figures 6 and 8, has been put in section 5.2.

At last, we moved lines 300 - 317 and all the parts referring to the description of the chosen simulated variables to the section 3 (Material and Methods).

**C7:** Lines 205 – 208: the authors write 'at millennial-scale, large magnitude PP oscillations, are observed during the deglaciation (19–11 kyr BP), showing similar features than those found in the Greenland ice core  $\delta$ 180 record, representing the rapid climatic changes in north hemispheric high-latitude areas (Fig. 2; Stuiver and Grootes, 2000).'

But it is stated in section 3.1 that the age model, although primarily based on 31 AMS 14C dates, it was still tuned to GISP2 Greenland ice core  $\delta$ 18O curve. Can this be considered circular reasoning?

**R7:** We thank Reviewer # 1 for highlighting this peculiar aspect. Indeed, it might be seen as a circular reasoning. However, our micropalaeontological data are well in phase with numerous geochemical data obtained elsewhere in the Tropical Indian Ocean and the Chinese continent, based on sediment cores and speleothems with totally independent age models, respectively. They also match very well TraCE-21 and IPSL-CM5A-LR outputs. Besides, as mentioned above (Reply 5), the age model of core MD77-176 has already been used by Marzin et al., (2013), Yu et al. (2018), and Ma et al. (2019), i.e. papers discussing geochemical data at regional and global scales. All these highlights point to a robust age model and demonstrate that our micropalaeontological data can-be properly be discussed in the light of the rapid climatic changes recorded in the northern high latitudes. To avoid any confusion, we rephrased this part of the manuscript focusing on the relationship that exists between PP and SSS of MD77-176.

**C8:** Lines 255 – 229: the authors write, 'Several pieces of evidence suggest that millennial- scale variations of PP between 26 and 19 kyr (i.e. before the LGM) chiefly resulted from wind-driven mixing. First, high PP values are reached during intervals of low surface water salinity. If these PP variations (and upper water column stratification) were primarily driven by precipitation—evaporation changes, the opposite relationship would be expected, and PP would peak at periods of higher salinity because of the weaker barrier layer effect'.

- a) Can you independently verify if the wind-driven mixing in the Northern Indian Ocean was enhanced during the LGM?
- b) Which are the intervals of low salinity during the LGM?
- c) Isn't the LGM Andaman Sea significantly more saline compared to other periods such as the Holocene?
- d) How does precipitation minus evaporation impact PP variability in general?
- e) What inferences can be made on LGM PP variability from the LGM experiments?

**R8:** We appreciate these remarks that rise further questions and clearly help us improving our interpretations. We first answer your questions one by one and then develop a more detailed response that echoes question a–e.

a) We checked the modeling outputs of surface winds during the both monsoonal seasons. It shows stronger summer wind and weaker winter wind intensities over the Bay of Bengal and Andaman Sea during the LGM (Fig. AC6new Fig. 7i, j).

b) The short intervals of low salinity are shown by the SSS record of MD77-176. They are recorded at  $\sim$ 21 kyr BP and  $\sim$ 23 kyr BP (Fig. AC4new Fig. 6h). However, it is not possible to test such specific short-term intervals with model outputs that give mean states of chosen parameters during the LGM.

c) The modeling outputs show that generally, Bay of Bengal and Andaman Sea behave the same way. That is only in the northeastern Bay of Bengal, close to the coasts, that a significant difference may be seen. Indeed, according to these model outputs, they are both getting saltier during the LGM, while the northeastern BoB is unchanged or a little fresher (Fig. AC5new Fig. 7h). The Andaman Sea doesn't appear specifically more saline than the BoB during that time interval.

d) According to IPSL-CM5A-LR outputs, it appears that if the net precipitation is lower during the LGM, the Bay of Bengal and Andaman Sea might get saltier and PP might increase due to weaker salinity stratification.

e) The LGMc experiment gives a mean state of PP during LGM. Generally, it shows higher PP in the BoB and the Andaman Sea. Under weaken AMOC condition, LGMf experiment shows higher PP compared to LGMc matching our reconstructed PP results from the LGM to the Heinrich 1.

**General reply:**

During glacial times (26–19 kyrs), high (low) PP intervals do match low (high) SSS ones, as shown by low (high) values in seawater oxygen anomalies recorded at the same site (Marzin et al., 2013; Fig. AC4new Fig. 6h, i).

There is no doubt that the South Asia and the North Indian Ocean are drier during the LGM due to relatively lower precipitation over the South Asia, as demonstrated by previous empirical data (Dutt et al., 2015; Contreras-Rosales et al., 2014; Kudrass et al., 2001) as well as numerical outputs here (Figs. AC4, AC5new Fig. 7f, g). However, the outputs of IPSL-CM5A-LR simulations, together with TraCE-21 ones show that, compared to preindustrial, weaker winter winds, stronger summer winds, and saltier sea surface conditions, generally prevailed in the Bay of Bengal and the Andaman Sea during the LGM (LGMc in Fig. AC5 new Fig. 7). These results suggest that the interpretation we have made for the last deglaciation and the Holocene, stating that a stronger summer monsoon and/or a weaker winter monsoon, induce increased precipitation, decreased SSS and thus, stronger salinity stratification and subdued PP is not always verified, and particularly during the LGM. In such a case, we cannot exclude that stronger and drier summer winds during that time interval (as suggested by model here), could eventually lead to enhanced sea-surface mixing, thus triggering upper water mixing, higher SSS, and higher PP as observed in the Arabian Sea today. However, as mentioned in the introduction of our manuscript, the Arabian Sea behave in a very different way than the Bay of Bengal, notably thanks to the development of massive upwelling on its western coasts, and the direct comparison of both basins may be questioned. Unfortunately, we cannot test such sea-surface mixing hypothesis with TraCE-21 or IPSL-CM5A-LR outputs, so far.

Spatial discrepancies of SSS are also found with model outputs. This is particularly the case when dealing with the northeastern Bay of Bengal and northern Andaman Sea areas. First, models in PMIP3 (Braconnot et al., 2012) show different results of SSS for the LGM: some models show fresher water, while others depict saltier conditions (Fig. AC10). Second, when dealing with the outputs of IPSL-CM5A-LR, such area (that include our core site) has very limited SSS increases during the LGM, if it doesn't show sometimes SSS decreasing trends (Fig. AC5new Fig. 7g). Such discrepancies have also been reported once by empirical data. Indeed, Sijinkumar et al. (2016) depict lower SSS in the northern

Andaman Sea during the LGM, i.e. under lower sea-level conditions. It may highlight the complex area that is the northeastern Bay of Bengal and northern Andaman Sea due to the Irrawaddy mouth influence. It might also partly explain the millennial-scale relationship documented at our core between SSS and PP at that time, i.e. under relatively low sea-level when site MD77-176 is located in a more proximal environment. Indeed, one cannot exclude that under such conditions, the PP increases (decreases) observed when SSS decreases (increases), reflect an increases (decreases) of nutrient together with freshwater inputs from the Irrawaddy river, respectively. Such assumption is confirmed in Figures AC2 and AC3new Fig. 3g —where PP strongly increases (Fig. AC2), and in the new vertical oceanic profiles we provide (Fig. 5e,j), wheren increased PP are accompanied by increased nutrient in surface layers thanks to a more proximal environment at our studied site highlighting a change vertical profiles clearly change from an open ocean type to a more coastal one one (Fig. AC3). Our scenario appears therefore to be a suitable explanation for the PP pattern obtained herein during the LGM.

However, in all cases, it seems difficult at that point, to deeply compare thoroughly (and discuss) the millennial PP changes obtained at our core site, to mean state simulations of local PP and SSS, obtained for the northern Bay of Bengal and Andaman Sea during the LGM. Additional high-resolution PP records and further numerical simulations are required in the area, in order to discuss this issue properly. As an example, a PP record further south in the Andaman Sea, i.e. far away from river mouth influences, (Zhou et al., unpublished) clearly shows higher PP from 30 to 19 ka, under saltier conditions, and does not show strong short-term fluctuations as recorded at site MD77-176.

The influence of drier and stronger summer winds together with the influence of nutrient and freshwater inputs from the Irrawaddy river behind PP variability during the LGM, are therefore evoked in the manuscript, but with extreme caution. In conclusion, we now interpret the reconstructed PP variations observed at site MD77-176 during the last glacial as the result of nutrient conditions changes within the upper layers, thanks to both, lower sea-level and enhanced influence of the Irrawaddy-Salween river mouth system.

**C9:** In section 4.1 (line 217): The authors write that, 'PP peaks are related to low SSS intervals before the LGM, and high SSS intervals over the last 19 kyr'.**

Although, PP did not significantly change over the course of the Holocene, there appears to be a clear discrepancy between the gradual monsoon intensification over the Holocene and PP variability. PP variability over the course of the last deglaciation and the Holocene are clearly different. Proxy data shown in Figure 2 suggest that estimated PP has lower valued during the Mid-Holocene (~90 gC m-2 yr-1) compared to late – Holocene (~130 gC m-2 yr-1). This, however, is not discussed in any detail and the way the discussion section is structured is at fault again.

**R9:** We agree with Reviewer #1. While the mechanisms controlling PP variations during the last deglaciation and the Holocene are similar and related to salinity stratification, PP variability is different over these two time intervals. They are characterized by rapid and large amplitude PP changes during the deglaciation, and rather gradual PP trends during the Holocene. Both periods are under the influence of insolation and AMOC forcing that impact land-sea thermal distribution over low latitudes, thus moderating monsoon strength, and controlling oceanic stratification and PP. However, to the different of the Holocene, rapid changes occur in the AMOC strength during the deglaciation, and they are clearly reflected in the Indian monsoon and PP dynamics at that time.

Therefore, such different PP patterns between the last deglaciation and the Holocene is clearly related to AMOC vs insolation imprints other the last 19 kyrs. Rapid changes in PP patterns during the last deglaciation clearly reflect the rapid changes in the AMOC strength. To the opposite, long-term changes in PP during the Holocene most probably reflect long-term changes in insolation and associated feedbacks with the ocean-atmosphere system. We now discuss the deglacial and Holocene PP variabilities separately, in our revised sections 5.2 and 5.3, respectively.

**C10:** In section 4.1 (lines 204 – 205): it is briefly mentioned that PP variability shows 'an opposite trend compared to insolation (Fig. 4a, h)' and in section 4.3.1 it is stated that 'insolation is the main climate forcing factor during the Holocene'.**

Why do we have the monsoon peaking later during the mid-Holocene lagging maximum Northern Hemisphere summer insolation by few kyrs then? The lagged response of the monsoon to insolation forcing suggests that orbital scale monsoon variability is more complex (see Clemens et al., 2003; Caley et al., 2011; Gebregiorgis et al., 2018). Having this in mind, I would therefore encourage the authors to have a more critical outlook on PP variability over the course of the Holocene. I also recommend including Figure S3 – S6 in the main text body and can be used to gain some unique insights on LGM, deglacial and Holocene PP variability. Perhaps Fig. 3 can be moved to the supplementary section.

**R10:** During the Holocene, our PP record shows a minimum at ~6–8ka, lagging of about few centuries3.5–5.5 kyr, the maximum North Hemisphere august insolation curve. However, it is clearly in phase with geochemical records obtained in the area that document high PP in the Arabian sea (Schulz et al., 1998; Ivanochko et al., 2005) and high precipitation over South Asia during that time interval (Dutt et al., 2015; Contreras-Rosales et al., 2014; Fig. AC4new Fig. 6). Such results show that during the Holocene, PP from the northeastern Bay of Bengal is highly related to monsoonal dynamic, and more particularly, precipitation. Summer winds triggers strong coastal upwelling and high PP in the Arabian Sea. They also transport moisture to the South Asia where the summer precipitation is strong. Such increase in precipitation causes strong salinity stratification over the northeastern Bay of Bengal and thus low PP.

The references cited in reviewer's comment argue for the hypothesis that tropical monsoon variability is dominated by, and responds directly to the North Hemisphere summer solar radiation, and point out the importance of internal climate forcing and oceanic feedbacks, such as latent heat export from the southern Indian Ocean. Clemens et al., (2003) particularly point out that the minima of SST in the southern subtropical Indian Ocean are synchronous with the maxima of summer monsoon, and the moderating effect of ocean thermodynamic features on monsoon circulation is important. In all cases this aspect is an (usually) inexplicable issue. We have mentioned this lag in the revised section 5.3 of the manuscript, and interpret the Holocene period with caution. The modifications of supplementary figures are explained in Reply 25.

**C11:** Line 57: What 'fast changes'? Please rephrase.**

R11: We have rephrased to 'abrupt changes'

C12: Line 61: PP record or paleo-PP record. Stick with one for consistency.

R12: We sticked with 'PP record'.

**C13:** Line 61: Da Silva et al., 2017 is a relevant reference here.

R13: We have cited this.

**C14:** Line 62: 'tropical ocean ecology' is very broad and I am not sure this is accurate as well. Perhaps Northern Indian Ocean ecology is more appropriate.

R14: We agree with this suggestion and made the changes in the light of the comment.

**C15:** Line 73: 'High-time-resolution' or 'High-temporal-resolution'? 'High-resolution' is a perhaps a better phrase.

R15: We have rephrased to 'high-resolution'

**C16:** Line 74: Why is it important that the 'studied period covers a complete precession cycle'? This sentences need to be qualified or delete otherwise.

**R16:** We've removed this sentence.

**C17:** Line 80: 'interpret' is perhaps a better word here than 'analyze'.

R17: We agree with this suggestion and made the changes when necessary.

**C18:** Line 199–200: 'At orbital scale' – remove.

R18: It has been done.

**C19:** Line 202: use 'maximum or minimum Northern Hemisphere (NH) summer insolation' instead of low or high insolation with no reference to the latitude or the season.

**R19:** It has been done.

C20: Line 205: 'On millennial timescale...'

**R20:** It has been done.

**C21:** Line 211: 'Synchronous vs. asynchronous' rather than 'Negatively vs. positively correlated' and of course 'correlation' being a statistical term.

R21: We agree with this suggestion and made the changes in the light of the comment.

*C22:* Line 290–291: Rephrase or remove

'During the Holocene, insolation is the main climate forcing factor since other forcing (i.e. greenhouse gas, ice volume, coastlines, vegetation) are relatively stable after the deglaciation.

**R22:** We have removed this sentence.

**C23:** Line 291–292: Rephrase. Perhaps, a sentence along these lines will do: 'the response of the Indian monsoon to changes in orbital insolation has previously been examined using both AGCMs and ocean–atmosphere general circulation models. . .(Refs).)'

'The mechanisms that force monsoon climate to change were studied by many modeling works (Refs).'

**R23:** We have removed this part, because of the new manuscript structure.

**C24:** Figures S1, S5, S6 are not cited in the main text and please add supplementary text to the supplementary information. Also make sure that the figures are in chronological order.

**R24:** We decided to keep all of the maps of TraCE-21 outputs in the Supplementary Material, and show all ofonly show the maps from our of IPSL-CM5A-LR results in the main text. They have been slightly modified to match the new structure/discussion of the manuscript. All of the figures presented in the Supplementary Material are summarized, within detailed captions.

**R25 (modification of supplementary figures 3 to 6):**

1) Fig. S3 have been modified and moved to the main text (Figs. AC5, AC6, AC8, AC9) The figure showing IPSL-CM5A results have been merged and moved to the main text (new Fig. 8). We show four groups of maps, which are the CTRL results as well as the differences between LGMc and CTRL, LGMf and LGMc, MH and CTRL. The variables are annual net precipitation (precipitation minus evaporation), annual SSS, annual potential gradient between 200 and 5 m, JJA surface wind speed and DJF surface wind speed.

2) For Figs. S4 to S6, we have removed the results of ORB simulation as they are similar to the FULL simulation, and removed the results of MWF\_BA minus MWF\_HS1 as well, since they are similar to the results of TraCE\_LGM minus MWF\_HS1. Therefore, for the maps of TraCE-21 simulations, we show five groups of maps in the Supplementary Material which are the LH, and differences between MH and LH, LGM and LH, BA and HS1, MWF\_HS1 between LGM. We show the same variables as IPSL-CM5A-LR. The modified supplementary figures can be seen from Fig. AC11 to AC16. We have removed the maps of TraCE-21 as they are similar to the IPSL-CM5A results.

**References**

Anderson, D. M. and Prell, W. L, 1992. The structure of the southwest monsoon winds over the

Arabian Sea during the late Quaternary: observation, simulations, and marine geologic evidence, Journal of Geophysical Research, 97, 15481–15487.

Bartolacci and Luther, 1999. Patterns of co-variability between physical and biological parameters in the Arabian Sea. Deep-Sea Research II, 46, 1933–1964.

**Braconnot et al., 2012**. Evaluation of climate models using palaeoclimatic data, Nature Climate Change, 2, 417–424.

**Clemens and Prell, 2003.** A 350,000 year summer-monsoon multi-prosy stack from the Owen Ridge, North Arabian Sea. Marine Geology, 201, 35–51.

**Contreras-Rosales et al., 2014.** Evolution of Indian Summer Monsoon and terrestrial vegetation in the Bengal region during the past 18 ka. Quaternary Science Reviews, 102, 133-148.

**Dutt et al., 2015.** Abrupt changes in Indian summer monsoon strength during 33,800 to 5500 years B.P. Geophysical Research Letter, 42, 5526-5532.

**Gardner et al., 1999.** The role of seasonal and diel changes in mixed-layer depth on carbon and chlorophyll distributions in the Arabian Sea, Deep-Sea Research II, 46, 1833–1858.

**Ivanochko et al., 2005.** Variations in tropical convection as an amplifier of global climate change at the millennial scale. Earth and Planetary Science Letter, 235, 302-314.

**Kudrass et al., 2001.** Modulation and amplification of climatic changes in the Northern Hemisphere by the Indian summer monsoon during the past 80 k.y. Geology, 29, 63-66.

**Ma et al., 2019.** Changes in Intermediate Circulation in the Bay of Bengal since the Last Glacial Maximum as inferred from benthic foraminifera assemblages and geochemical proxies. Geochemistry, Geophysics, Geosystems, 20, 1592-1608.

Madhupratap et al., 1996. Mechanism of the biological response to winter cooling in the northeastern Arabian Sea, Nature, 384, 549–552.

Liao et al., 2016. Potential new production in two upwelling regions of the western Arabian Sea: Estimation and comparison. Journal of Geophysical Research: Oceans, 121, 4487–4502.

**Marzin et al., 2013.** Glacial fluctuations of the Indian monsoon and their relationship with North Atlantic climate: new data and modeling experiments, Climate of the Past, 9, 2135–2151.

Schulz et al., 1998. Correlation between Arabian Sea and Greenland climate oscillations of the past 110,000 years. Nature, 393, 54-57.

Schneider et al., 2014. Migrations and dynamics of the intertropical convergence zone. Nature, 513, 45–53.

**Sijinkumar et al., 2016.** δ18O and salinity variability from the Last Glacial Maximum to Recent un the Bay of Bengal and Andaman Sea. Quaternary Science Reviews, 135, 79–91.

Wiggert et al., 2005. Monsoon-driven biogeochemical processes in the Arabian Sea. Progress in Oceanography, 65, 176–213.

**Yu et al., 2018.** Antarctic Intermediate Water penetration into the Northern Indian Ocean during the last deglaciation. Earth and Planetary Science Letters, 500, 67–75.

Figures

Fig. AC1. Relative abundance changes of main coccolith species and reconstructed PP.